# iPSC Technology Revolutionizes CAR-T Cell Therapy for Cancer Treatment

**DOI:** 10.3390/bioengineering12010060

**Published:** 2025-01-13

**Authors:** Jiepu Zong, Yan-Ruide Li

**Affiliations:** Department of Microbiology, Immunology & Molecular Genetics, University of California, Los Angeles, CA 90095, USA

**Keywords:** induced pluripotent stem cell (iPSC), chimeric antigen receptor (CAR)-engineered T (CAR-T) cells, cancer therapy, allogeneic cell therapy, genetic engineering, graft-versus-host disease (GvHD), T cell receptor (TCR)

## Abstract

Chimeric Antigen Receptor (CAR)-engineered T (CAR-T) cell therapy represents a highly promising modality within the domain of cancer treatment. CAR-T cell therapy has demonstrated notable efficacy in the treatment of hematological malignancies, solid tumors, and various infectious diseases. However, current CAR-T cell therapy is autologous, which presents challenges related to high costs, time-consuming manufacturing processes, and the necessity for careful patient selection. A potential resolution to this restriction could be found by synergizing CAR-T technology with the induced pluripotent stem cell (iPSC) technology. iPSC technology has the inherent capability to furnish an inexhaustible reservoir of T cell resources. Experimental evidence has demonstrated the successful generation of various human CAR-T cells using iPSC technology, showcasing high yield, purity, robustness, and promising tumor-killing efficacy. Importantly, this technology enables the production of clinical-grade CAR-T cells, significantly reducing manufacturing costs and time, and facilitating their use as allogeneic cell therapies to treat multiple cancer patients simultaneously. In this review, we aim to elucidate essential facets of current cancer therapy, delineate its utility, enumerate its advantages and drawbacks, and offer an in-depth evaluation of a novel and pragmatic approach to cancer treatment.

## 1. Introduction

Chimeric antigen receptor (CAR)-engineered T (CAR-T) cell therapy has emerged as a transformative approach in cancer treatment, offering promising therapeutic potential [1,2,3]. This strategy involves the genetic engineering of T cells to express CARs, which enable precise targeting of tumor-specific antigens on the surface of malignant cells. To date, the U.S. Food and Drug Administration (FDA) has approved seven CAR-T cell products, each of which has contributed to significant clinical outcomes. In particular, CAR-T cell therapies have markedly improved five-year survival rates, increasing them from 60% to approximately 90% in specific hematological malignancies. However, the high cost of these therapies, reaching approximately $275,000 per treatment as of 2022 [4,5,6], presents a substantial financial barrier to widespread use. Notably, various preclinical studies have investigated the application of CAR-T cell therapy in other cancers, including silent solid tumors, such as employing mesothelin-targeting CAR-T cells for the treatment of triple-negative breast cancer [7]. However, the efficacy of CAR-T cell therapy in solid tumors remains constrained due to challenges related to poor tumor homing and infiltration.

Adverse effects, such as headache, myalgia, and pyrexia, are common during treatment, underscoring the need for careful management of patient outcomes. Autologous CAR-T cells, which are generated from a patient’s own T cells, present challenges in terms of scalability, particularly due to the limited availability of T cells for redosing (Figure 1) [8,9,10]. In contrast, allogeneic CAR-T cells, derived from healthy donors, offer the potential for an inexhaustible supply of therapeutic cells. However, these cells tend to exhibit reduced persistence in vivo. Additionally, the use of allogeneic cells introduces the risk of graft-versus-host disease (GvHD), where the donor-derived cells may elicit an immune response against the recipient’s tissues [5,11]. To overcome these challenges, the development of advanced technologies for optimizing allogeneic CAR-T cell therapies is crucial.

The clinical potential of T cell therapy is hindered by the scarcity of antigen-specific human T lymphocytes, necessitating innovative strategies to enhance T cell availability and applicability. Exploiting a robust source of T cells, such as those derived from pluripotent stem cells (PSCs), holds promise. Pluripotent stem cells can be employed to generate an ample supply of functional T cells, addressing this limitation. Notably, peripheral blood T lymphocytes obtained from a healthy donor can be transduced to establish T cell-derived induced pluripotent stem cell clones (T-iPSCs) [12,13,14]. By leveraging the differentiating capabilities of iPSC technology, immune cells, including T cells, can be generated. A notable advancement is the development of iPSC-derived EZ-T cells, mimicking the differentiation process of naive T cells into effector cells and memory-like T cell subsets. Recent studies have successfully generated allogeneic CAR-T cells from iPSCs, exemplified by 1928z-T-iPSC-T, CAR iPSC-T cells, and T-iPSCs [14,15,16]. These investigations underscore the feasibility, safety, and therapeutic efficacy of iPSC-derived CAR-T cells. However, existing challenges remain. The conversion of differentiated cells into iPSCs remains inefficient, impeding the process. Moreover, the existence of numerous iPSC lines lacking authentic pluripotency further compounds the issue, as they exhibit inadequate differentiation potential across essential embryonic cell lineages. Addressing these limitations is pivotal to fully harnessing the potential of iPSC-based T cell therapies.

In this review, we explore the fundamental aspects and applications of current CAR-T therapy, highlighting its advantages and limitations (Figure 1). Additionally, we provide an overview of the latest iPSC-derived CAR-T cell therapy platforms (Table 1).

## 2. Allogeneic CAR-T Cell Therapy

### 2.1. Conventional Autologous CAR-T Cell Therapy

Currently, CAR-T cell-based therapy has shown promise in a broad range of cancers, including blood cancers and solid tumors. In particular, CAR-T cell therapy has shown remarkable efficacy in treating B cell Acute Lymphoblastic Leukemia (B-ALL) and multiple myeloma (MM) [17,18,19]. CAR-T therapy was able to treat 90% of patients having B-ALL, and 80% of patients having MM. In addition, more CAR-T therapies targeting other cancers are under preclinical and clinical studies, such as mesothelin-targeting CAR-T cells to treat ovarian cancer, and CD70-targeting CAR-T cells to treat acute myeloid leukemia (AML) [20,21]. Currently, multiple generations of CAR-T cell therapy have been developed. First-generation CARs are composite proteins composed of an external domain that binds to antigens (often the single-chain variable fragment of an antibody) linked with an internal signaling domain—frequently the CD3ζ chain from the TCR. In second-generation CARs, the effectiveness of CAR-T cell action is heightened by incorporating a costimulatory domain joined with CD3ζ, like CD28 or 4-1BB. This fusion aims to bolster the expansion and longevity of engineered cells within the body. Third-generation CARs, featuring multiple costimulatory domains, have also been successfully devised [22].

Autologous second-generation CAR-T cells have achieved repeated complete remissions in patients with hematological malignancies previously considered untreatable. This success has resulted in the approval of two therapies, tisagenlecleucel and axicabtagene ciloleucel, for the treatment of relapsed or refractory B cell acute lymphoblastic leukemia, diffuse large B cell lymphoma, and primary mediastinal large B cell lymphoma. As a result, CAR-T cell therapy represents a significant milestone in medical progress [23,24,25].

Presently, all existing CAR-T therapies are autologous in nature, implying the utilization of patient-derived donor cells. This characteristic attribute has led to a constraint in the frequency of re-administration due to the finite availability of such cells. Consequently, the comprehensive treatment of all individuals afflicted with cancer becomes an arduous task. The manufacturing process itself poses formidable challenges, encompassing intricate logistical hurdles, delays spanning from leukapheresis to CAR-T cell infusion, heterogeneity in T cell profiles influenced by individual immune attributes, and the repercussions of preceding therapeutic interventions. Furthermore, the economic outlay associated with the autologous modality is notably elevated, potentially reaching up to $373,000 per treatment [26,27].

In an endeavor to surmount these challenges, allogeneic CAR-T cells emerge as an imperative alternative [24,28]. By employing allogeneic CAR-T cells, the prospect of simultaneously treating multiple patients becomes feasible. These donor cells are sourced from healthy individuals, thus ensuring a practically limitless cell reservoir. The manufacturing process for allogeneic CAR-T cells is streamlined and more productive, embracing an industrially scaled-up protocol that yields a substantial quantity of CAR-T cells from a solitary donor, amenable to cryopreservation. Additionally, the financial outlay associated with allogeneic CAR-T therapy is notably diminished when contrasted with its autologous counterpart [5,29,30].

Nonetheless, notwithstanding its merits, allogeneic CAR-T therapy is not devoid of significant risks and challenges. The foremost concern pertains to graft-versus-host disease (GvHD), an outcome of the immune rejection of donor cells from a disparate healthy individual. This precipitates a cascade of immunological responses within the recipient’s body. The etiology of GvHD stems from mismatches within the major histocompatibility complex (MHC) of the donor and recipient cells. Various strategies, such as TCR disruption, have been developed to mitigate the incidence and severity of GvHD [31,32,33]. Furthermore, allogeneic CAR-T therapy’s durability tends to be short to intermediate. In contrast, the effect of autologous interventions tends to exhibit a more sustained timeframe, ranging from months to weeks.

### 2.2. Allogeneic TCR-Disrupted CAR-T Cell Therapy

A novel therapeutic approach termed allogeneic TCR-disrupted CAR-T cell therapy has exhibited notable advantages over conventional iterations. TCRs and CARs are prominent tools in the realm of stem cell engineering, accentuating immune cell specificity. Utilizing either peripheral blood mononuclear cell (PBMC)-derived or stem cell-derived immune cells, coupled with retroviral or lentiviral vectors, TCRs and CARs can be durably integrated into stem cells, furnishing a long-term therapeutic strategy. This addresses a significant limitation of conventional allogeneic CAR-T therapy, which is characterized by transitory efficacy.

Furthermore, TCR-disrupted CAR-T cells have the capacity to mitigate or circumvent GvHD attributed to allogeneic CAR-T therapy. GvHD, facilitated by alloreactive donor T cells, can be averted through TCR manipulation, involving editing or employing immune cells with established specificities or unresponsiveness to peptide-MHC disparities. The ablation of TCR gene function signifies a preclusion of T cell-mediated GvHD induction. For instance, when TCR TRAC is knocked out, host immune recognition is abrogated, forestalling subsequent reactions, notably GvHD initiation [34,35,36].

To expand, targeting gene knockouts such as B2M and CIITA can forestall host T cell-mediated allorejection. Additionally, the targeted knockout of immune checkpoint genes (e.g., PDCD1, *LAG3*, *CTLA4*, and *DGKa*) augments immune cell-mediated antitumor efficacy, potentiating their tumor-suppressive capacity [30,31]. This genetic modulation enhances the immune cells’ ability to counteract tumors.

Implementing CRISPR-Cas9 technology, the integration of CAR genes into the *TRAC* locus engenders uniform CAR expression across T cells while concurrently eliminating endogenous TCR expression, thereby heightening T cell effectiveness [31,32,33]. Notably, CRISPR-Cas9-mediated genome-wide screening of therapeutic immune cells offers a valuable strategy to identify gene targets pertinent to cell-based therapeutic interventions.

It is important to note that host cell-mediated allorejection poses a significant concern for allogeneic CAR-T cell therapy, particularly with respect to host T cell and NK cell-mediated allorejection [37]. Mitigating allorejection is critical for ensuring the long-term sustainability and efficacy of off-the-shelf cell products. Various strategies have been employed to enable allogeneic CAR-T cells to evade host cell recognition, including ablation of HLA molecules, overexpression of NK cell inhibitory ligands, overexpression of anti-apoptotic genes, immunomodulation, utilization of stem cell technology to produce cells with low alloreactivity, and HLA matching along with careful donor selection [5,37,38]. To enhance the persistence and efficacy of allogeneic CAR-T cells, lymphodepletion using agents such as alemtuzumab, fludarabine, and cyclophosphamide is necessary [39,40,41].

### 2.3. Limitations of Current CAR-T Cell Therapy

The promising potential of intricate genetic modifications for complex editing is tempered by potential drawbacks, including increased aneuploidy frequency, heightened risk of tumorigenicity, and compromised in vivo efficacy against tumors, along with reduced persistence of therapeutic cells. Recent research underscores the importance of maintaining TCR expression in CAR-T cell therapies. This is exemplified by compromised cytokine production and diminished long-term persistence observed in TRAC-knockout CAR19-T cells in comparison to CAR19-T cells retaining intact TCRs. Consequently, innovative strategies are imperative for advancing allogeneic CAR-T cell development, particularly leveraging stem cell-derived methodologies such as induced pluripotent stem cells (iPSCs) to surmount these challenges and enhance allogeneic CAR-T therapy efficacy [12,16,42].

An additional constraint arises from the toxicity associated with lymphodepletion essential prior to each CAR-T cell administration. To address this, it becomes essential to optimize allogeneic CAR-T cells by restricting lymphodepletion intensity and consolidation cycles. Moreover, limitations pertain to the feasibility of redosing due to repeated lymphodepleting conditioning before CAR-T cell infusion, primarily due to linked toxicities. This concern arises from the need for moderate-intensity conditioning regimens to facilitate CAR-T cell homeostatic expansion. Overcoming this limitation could necessitate employing next-generation allogeneic cells engineered for host immune system evasion. Finally, the restricted availability of these cells is a potential impediment, demanding extensive ex vivo expansion strategies.

## 3. iPSC-Derived Allogeneic CAR-T Cell Products

### 3.1. Generate iPSC-Derived CAR-T Cells Using an OP9-DL1 Feeder-Dependent Culture

A previous study successfully developed iPSC-derived CAR-T cells targeting CD19 to treat B cell malignancies and infectious diseases [14]. This work highlighted a significant limitation in T cell therapies: the restricted availability of T cells. Conventional adoptive T cell therapies require either labor-intensive generation of T cell lines from carefully selected donors or genetic modification of autologous T cells from each patient, complicating the broad application of therapies with predefined antigen specificity. By contrast, iPSC-derived CAR-T cells, which have shown robust tumor-killing abilities, offer a promising alternative. While pluripotent stem cells can provide an unlimited source of T cells, the full therapeutic potential of these iPSC-derived lymphoid cells has yet to be thoroughly understood.

In their methodology, the researchers combined iPSC and CAR engineering technologies to create human T cells that specifically target CD19, a common antigen on malignant B cells, in vitro [14]. These iPSC-derived CAR-T cells demonstrated a phenotype resembling innate γδ T cells [43,44]. Similarly to CAR-transduced γδ T cells sourced from peripheral blood, the iPSC-derived T cells showed strong tumor-suppressive effects in a xenograft model. The study suggests that genetic engineering of iPSCs with second-generation CARs could provide an effective approach to leverage the limitless supply of iPSCs, enabling the production of functional, phenotypically tailored T cells with precise antigen-targeting abilities.

To accomplish this, the researchers reprogrammed peripheral blood T lymphocytes from a healthy donor into iPSCs by introducing retroviral vectors carrying the transcription factors KLF4, SOX2, OCT4, and c-MYC [14]. This approach is highly significant, as it offers the potential to create an unlimited supply of T lymphocytes with targeted antigen specificity, bypassing the need for HLA matching. Leveraging the flexibility of pluripotent stem cells in combination with CAR technology, this method could also support the generation of diverse T cell subpopulations with additional genetic enhancements, thus expanding its applicability across various therapeutic fields.

### 3.2. Generate iPSC-Derived CAR-T Cells Using a Stroma-Free Culture

A recent study highlights the promise of iPSCs as a plentiful and adaptable resource for cell therapies [15]. This research demonstrates that inhibiting the *EZH1* gene significantly enhances the efficient differentiation and maturation of T cells from iPSCs in vitro. By combining EZH1 knockdown-induced epigenetic modification with a stroma-free T cell differentiation platform, researchers successfully generated iPSC-derived T cells, named EZ-T cells. These cells feature a highly diverse TCR repertoire and display molecular traits similar to peripheral blood-derived conventional αβ T cells. Upon activation, EZ-T cells develop into both effector and memory T cell subsets. Furthermore, introducing CARs into EZ-T cells enables them to exhibit robust anti-tumor activity in both in vitro tumor cell-killing assays and xenograft animal models [15].

This study presents an innovative stroma-free protocol for differentiating T cells from iPSCs that have been transduced with pre-engineered TCRs or derived from antigen-specific cytotoxic T cells. These iPSCs, harboring specialized TCRs, exhibit unique differentiation kinetics compared to their wild-type counterparts. To evaluate the efficiency and yield of the stroma-free approach, T cell differentiation was conducted using both this new method and a standard OP9-DL1 co-culture system. Significantly, the stroma-free protocol resulted in a marked increase in the generation of CD3^+^ T cells [15].

Prior studies have highlighted the regulatory influence of EZH1 in maintaining hematopoietic multipotency, where its suppression has been shown to promote lymphoid lineage potential during embryonic development in both mouse and zebrafish models [45,46]. Building on this knowledge, the current study investigates the role of EZH1 inhibition in facilitating in vitro T cell differentiation from iPSCs. This was achieved by employing shRNA-mediated knockdown of EZH1 during the differentiation of iPSC-derived CD34^+^ hematopoietic stem and progenitor cells into T cells [15]. Functional assays revealed that EZ-T cells exhibited enhanced effector properties, showing significantly higher CD69 expression upon PMA/ionomycin stimulation than either iPSC-OP9-T or iPSC-SF-T cells. Furthermore, EZ-T cells displayed increased CD107a expression compared to iPSC-SF-T controls under similar conditions. These findings contribute to the ongoing development of in vitro protocols for generating T cells from human pluripotent stem cells, guided by a deeper understanding of the key signaling pathways involved in T cell development.

### 3.3. Generating iPSC-Derived CAR-T Cells Using a 3D-Organoid Culture

A recent study explores the potential for generating an unlimited supply of CAR-T cells from human iPSCs to support off-the-shelf CAR-T cell immunotherapy [16]. The study addresses the challenges of efficiently differentiating iPSCs into conventional αβ T cell lineages while preserving both CAR expression and functionality. Researchers successfully reprogrammed iPSCs from CD62L^+^ naïve and memory T cells, introduced CD19-targeting CAR engineering, and achieved differentiation within a 3D-organoid culture [16]. This approach yielded iPSC-derived CD8αβ-positive CAR-T cells that closely resemble traditional CD8αβ-positive CAR-T cells. The expanded iPSC-derived CD19-CAR-T cells demonstrated comparable levels of antigen-specific activation, degranulation, cytotoxicity, and cytokine secretion to those observed in traditional CAR-T cells. Additionally, the iPSC-derived cells maintained stable expression of the TCR from the original T cell clone. In vivo studies further confirmed that iPSC CD19-CAR-T cells effectively mediated strong antitumor responses, extending the survival of mice bearing CD19-positive human tumor xenografts [16].

The study also emphasizes the constraints of current autologous CAR-T cell production, which depends on personalized blood apheresis and bespoke manufacturing steps. To overcome these challenges, the researchers utilized a 3D-organoid culture system to efficiently produce functional, mature human T cells from hematopoietic stem cells (HSCs) and embryonic stem cells (ESCs) [16,47,48,49].

The study further investigated the effector functions of reprogrammed iPSC-derived CD19-CAR-T cells in vitro, focusing on their capacity to target CD19-expressing cells. Additionally, it examined the signaling pathways activated in these CAR-T cells upon co-culture with CD19^+^ and CD19-knockout NALM6 cells. The results showed that iPSC CD19-CAR-T cells exhibited phosphorylation of ERK1/2 at Thr202/Thr204 and PLCγ at Ser1248 in a pattern similar to that of conventional CD19-targeting CAR-T cells, and in a manner specific to the presence of the CD19 antigen [16].

## 4. Discussion

Although iPSC-derived CAR-T cells present numerous advantages, they are still in the preclinical development stage and face several limitations. Strategies to enhance their anti-tumor efficacy remain an active area of research. Studies found that iPSC-derived CD19-targeting CAR-T cells and conventional CD8^+^ CAR-T cells demonstrated comparable anti-tumor activity across various tumor models. However, their effectiveness in prolonging survival in aggressive tumors was only modest. While CAR constructs incorporating cytokines, such as IL-15, have shown promise in boosting CAR-T cell potency, the engineered expression of IL-15 could potentially hinder iPSC differentiation into T cells. This is because IL-15 strongly promotes the differentiation of lymphoid progenitor cells into natural killer (NK) cells rather than T cells [50,51,52,53].

Furthermore, investigating the in vivo anti-tumor effects of unexpanded iPSC-derived CAR-T cells or those with elevated CAR expression levels is of interest, and such strategies are currently under investigation. Additionally, the iPSC-derived EZ-T cells, produced through established methodologies, predominantly comprise CD8 cytotoxic T cells. Consequently, innovative strategies are essential to enhance the efficient generation of mature CD4 single-positive T cells. A balanced ratio of cytotoxic to helper T cells has been associated with improved therapeutic outcomes [26,54].

The current strategies for attenuating the activity of the EZH1 enhancer during T cell differentiation primarily involve the use of viral vectors that integrate into the host genome. While dual inhibitors targeting EZH1 and EZH2 have demonstrated potential in promoting NK cell development, small molecule treatments have failed to replicate the effects achieved through short hairpin RNA (shRNA)-mediated knockdown of EZH1 during the differentiation of iPSC-derived T cells [15,55]. In light of the absence of EZH1-specific inhibitors that do not concurrently inhibit EZH2, the advancement of non-integrating gene knockdown approaches is imperative for translating EZH1-targeted T cell therapies into clinical practice. Additionally, genetic engineering approaches, such as the introduction of immune-enhancement genes like IL-15, IL-2, IL-7, and IL-18, should be explored [56,57].

Considering other cell carriers, such as NK and invariant natural killer T (iNKT) cells, could also be reasonable, as they induce fewer GvHD and other side effects. For instance, Li et al. reported a method for efficiently differentiating iPSCs into CAR-engineered NK cells exhibiting potent antitumor efficacy [58]. Notably, this study highlighted the design of a CAR that incorporates the transmembrane domain of NKG2D, the 2B4 co-stimulatory domain, and the CD3ζ signaling domain, which collectively mediate robust antigen-specific signaling in these iPSC-derived CAR-engineered NK cells [58]. The NK-like CAR demonstrated superior antitumor efficacy compared to conventional CARs that utilize CD28 or 4-1BB as co-stimulatory domains. However, further analyses are necessary to fully evaluate the implications of these findings.

Additionally, iPSC technology has been employed to generate CAR-engineered macrophages. Zhang et al. established a protocol for differentiating myeloid/macrophage lineages from CAR-iPSCs, thereby generating CAR macrophages that exhibited potent antitumor efficacy both in vitro and in vivo [59]. In another study, Abdin et al. reported a scalable method for generating functional CAR-engineered macrophages derived from iPSCs. These CAR-macrophages demonstrated enhanced, antigen-dependent phagocytosis of CD19^+^ target cancer cells, accompanied by increased pro-inflammatory responses [60]. Overall, the robustness of iPSC technology enables the generation of various types of CAR-engineered immune cells with high efficacy and safety, as well as specific tumor homing and targeting capabilities. This advancement provides significant opportunities for the treatment of a range of challenging solid tumors.

In addition to the differentiation in antitumor cytotoxic CAR-T cells, a recent study has highlighted the development of human iPSC-derived CAR-engineered CD4^+^ regulatory T cell (Treg)-like cells for the management of GvHD in a xenograft model [61]. The generated cells were induced with high levels of FOXP3 and demonstrated immunosuppressive functions, indicating their potential utility in suppressing the progression of GvHD as well as other diseases such as autoimmune disorders [61].

Similarly to conventional CAR-T cell therapy, iPSC-derived CAR-T cell products can be combined with various strategies to enhance their in vivo persistence and functionality. These strategies include radiotherapy, chemotherapy, monoclonal antibodies, oncolytic viruses, and cancer vaccines [62]. Exploring optimal combination therapies with iPSC-derived CAR-T cell therapy will be an essential avenue for future research, as it has the potential to maximize therapeutic efficacy in cancer treatment.

Generating T cells and CAR-T cells through extrathymic culture systems, whether single-layer or 3D-organoid co-cultures, remains challenging. Previously reported iPSC CAR-T cells generated using a monolayer coculture system exhibited an innate-like phenotype and demonstrated less efficient antigen-specific cytotoxicity and cytokine secretion compared to conventional CAR-T cells [63,64,65]. Developing a feeder-free system that supports efficient T cell differentiation and generation without the use of mouse-derived feeder cells is crucial for advancing clinical and translational applications [66]. This innovation not only enhances the safety and scalability of T cell therapies but also paves the way for more effective treatments, bringing us closer to realizing the full potential of immunotherapy in combating cancers and other diseases.

## Figures and Tables

**Figure 1 bioengineering-12-00060-f001:**
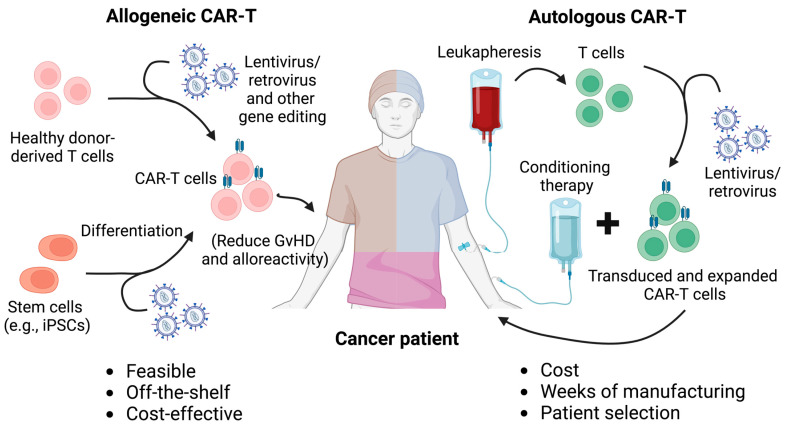
Comparison between autologous and allogeneic CAR-T cell therapy. Autologous and allogeneic CAR-T cell therapies differ primarily in their source of T cells and their implications for treatment logistics, safety, and efficacy. Autologous CAR-T cell therapy involves harvesting T cells from the patient’s own body, genetically engineering them to express a CAR, and then reinfusing them into the same patient. This personalized approach reduces the risk of immune rejection but presents challenges in terms of time, cost, and variability in T cell quality, especially in patients with weakened immune systems. In contrast, allogeneic CAR-T cell therapy uses T cells from healthy donors or stem cells such as iPSCs, which are engineered and prepared as an off-the-shelf product. This approach enables faster treatment delivery and the potential for mass production, making it more scalable. However, allogeneic CAR-T cells carry a higher risk of complications, such as graft-versus-host disease (GvHD) and immune rejection, which require additional genetic modifications, such as the disruption of the T cell receptor (TCR) to mitigate these risks. Additionally, strategies such as the ablation of HLA molecules and the overexpression of NK cell inhibitory ligands have been applied to allogeneic CAR-T cells to mitigate host cell-mediated allorejection and enhance the in vivo persistence and antitumor efficacy of the CAR-T cells.

**Table 1 bioengineering-12-00060-t001:** Comparison of three iPSC-derived CAR-T cell platforms.

iPSC-Derived CAR-T Cell Platforms	iPSC Source	iPSC Technology	Differentiation Approach	CAR Engineering	Characterization of iPSC-Derived CAR-T Cells	Limitations	Year and Reference
Platform 1	T cell-derived iPSCs (T-iPSCs)	Peripheral blood T lymphocytes are transduced with two retroviral vectors, each encoding two of the reprogramming factors KLF4, SOX2, OCT-4, and C-MYC	Mesoderm formation, hematopoietic specification and expansion, and T-lymphoid commitment (OP9-DL1 culture)	CD19-targeting CAR engineering on T-iPSCs	Display a phenotype resembling that of innate γδ T cells, and elicit strong anti-tumor responses in vivo	The generated iPSC-derived CAR-T cells have the properties of γδ T cells, and the OP9-DL1 culture involves murine-derived feeder cells	2013 [14]
Platform 2	CD62L^+^ naive and memory T cell-derived iPSCs	The T cells are transduced and reprogramed by episomal plasmids encoding KLF4, SOX2, OCT-4, C-MYC, and LIN28, along with P53 shRNA	Mesoderm induction, hematopoietic induction, T cell differentiation (3D-organoid culture), and expansion	CD19-targeting CAR engineering on iPSCs	Show antigen-specific activation, degranulation, cytotoxicity, and cytokine secretion, and mediate potent antitumor activity in vivo	The 3D-organoid culture involves murine-derived feeder cells	2022 [16]
Platform 3	Human erythroblast-derived iPSCs (cell line 1157)	NA	Embryoid body formation, EZH1 repression, and T cell differentiation (stroma-free culture)	CD19-targeting CAR engineering on iPSC-derived T cells	Give rise to memory-like T cells upon activation, and display enhanced antitumor activity in vitro and in vivo	iPSC-derived EZ-T cells predominantly consist of CD8 cytotoxic T cells, and a large amount of CAR viruses are utilized	2022 [15]

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
