# Peer review of "iPSC Technology Revolutionizes CAR-T Cell Therapy for Cancer Treatment"

_bioengineering, 2025, doi:10.3390/bioengineering12010060_

Round 1
Reviewer 1 Report
Comments and Suggestions for Authors
The review articles titled as iPSC Technology Revolutionizes CAR-T Cell Therapy for Cancer Treatment " effectively highlights the potential synergy between CAR-T cell therapy and iPSC technology as a promising strategy to address the limitations of CAR-T cell therapy.
I do believe it’s a topic of high relevance and importance to the readers of the journal and overall scientific community.
However, the review article lacks critical analysis, generate no unique discussion and is of insufficient scientific vigor.
Limited discussion of iPSC-Derived CAR-T Cells.
This reviewer also noticed increased self-citation. Therefore, discouraged to endorsed the publication of the review article.
Best wishes…
Author Response
The review articles titled as iPSC Technology Revolutionizes CAR-T Cell Therapy for Cancer Treatment " effectively highlights the potential synergy between CAR-T cell therapy and iPSC technology as a promising strategy to address the limitations of CAR-T cell therapy.
I do believe it’s a topic of high relevance and importance to the readers of the journal and overall scientific community.
However, the review article lacks critical analysis, generate no unique discussion and is of insufficient scientific vigor.
Limited discussion of iPSC-Derived CAR-T Cells.
This reviewer also noticed increased self-citation. Therefore, discouraged to endorsed the publication of the review article.
Best wishes…
Response: We appreciate the reviewer’s comments and would like to address them by incorporating a more critical analysis and discussion regarding iPSC-derived CAR-T cells. Specifically, we have included the following discussion in the revised manuscript:
“Considering other cell carriers, such as NK and invariant natural killer T (iNKT) cells, could also be reasonable, as they induce fewer GvHD and other side effects. For instance, Li et al. reported a method for efficiently differentiating iPSCs into CAR-engineered NK cells exhibiting potent antitumor efficacy[58]. Notably, this study highlighted the design of a CAR that incorporates the transmembrane domain of NKG2D, the 2B4 co-stimulatory domain, and the CD3ζ signaling domain, which collectively mediate robust anti-gen-specific signaling in these iPSC-derived CAR-engineered NK cells[58]. The NK-like CAR demonstrated superior antitumor efficacy compared to conventional CARs that utilize CD28 or 4-1BB as co-stimulatory domains. However, further analyses are necessary to fully evaluate the implications of these findings.
Additionally, iPSC technology has been employed to generate CAR-engineered macrophages. Zhang et al. established a protocol for differentiating myeloid/macrophage lineages from CAR-iPSCs, thereby generating CAR-macrophages that exhibited potent antitumor efficacy both in vitro and in vivo[59]. In another study, Abdin et al. reported a scalable method for generating functional CAR-engineered macrophages derived from iPSCs. These CAR-macrophages demonstrated enhanced, antigen-dependent phagocy-tosis of CD19+ target cancer cells, accompanied by increased pro-inflammatory respons-es[60]. Overall, the robustness of iPSC technology enables the generation of various types of CAR-engineered immune cells with high efficacy and safety, as well as specific tumor homing and targeting capabilities. This advancement provides significant opportunities for the treatment of a range of challenging solid tumors.
In addition to the differentiation into antitumor cytotoxic CAR-T cells, a recent study has highlighted the development of human iPSC-derived CAR-engineered CD4+ regu-latory T cell (Treg)-like cells for the management of GvHD in a xenograft model[61]. The generated cells were induced with high levels of FOXP3 and demonstrated immuno-suppressive functions, indicating their potential utility in suppressing the progression of GvHD as well as other diseases such as autoimmune disorders[61].
Similar to conventional CAR-T cell therapy, iPSC-derived CAR-T cell products can be combined with various strategies to enhance their in vivo persistence and functionality. These strategies include radiotherapy, chemotherapy, monoclonal antibodies, oncolytic viruses, and cancer vaccines[62]. Exploring optimal combination therapies with iPSC-derived CAR-T cell therapy will be an essential avenue for future research, as it has the potential to maximize therapeutic efficacy in cancer treatment.”
Reviewer 2 Report
Comments and Suggestions for Authors
A state-of-the-art review of one of the most advanced cancer treatments.
In general, the topic is covered quite fully, but there are shortcomings and some sections are not sufficiently covered.
The comments are as follows:
1. Page 2. Recent studies have successfully generated allogeneic CAR-T cells from iPSCs, exemplified by 1928z-T-iPSC-T, CAR-iPSC T cells, and T-iPS.
Only one reference, need more.
2. I didn’t find these examples (1928z-T-iPSC-T, CAR-iPSC T cells) in article Jing, R., Scarfo, I., Najia, M.A., Lummertz da Rocha, E., Han, A., Sanborn, M., Bingham, T., Kubaczka, C., Jha, D.K., Falchetti, M., et al. (2022). EZH1 repression generates mature iPSC-derived CAR T cells with enhanced antitumor activity. Cell Stem Cell 29, 1181-1196.e6. 10.1016/j.stem.2022.06.014
3. Page 4. Additionally, the financial outlay associated with allogeneic CAR-T therapy is notably diminished when contrasted with its autologous counterpart. Reference is required to o confirm that phrase.
4. Furthermore, the economic outlay associated with the autologous modality is notably elevated, potentially reaching up to $373,000 per treatment. Didn’t find about economic outlay in article Li, Y.-R., Dunn, Z.S., Yu, Y., Li, M., Wang, P., and Yang, L. (2023). Advancing cell-based cancer immunotherapy through stem cell engineering. Cell Stem Cell. 10.1016/j.stem.2023.02.009
5. Page 5. Various strategies, such as TCR disruption, have been developed to mitigate the incidence and severity of GVHD. Reference is required to o confirm that phrase. Correct the abbreviation to GvHD
6. Reference is required to o confirm that phrase: «Implementing CRISPR-Cas9 technology, the integration of CAR genes into the TRAC locus engenders uniform CAR expression across T cells while concurrently eliminating endogenous TCR expression, thereby heightening T cell effectiveness. Notably, CRISPRCas9-mediated genome-wide screening of therapeutic immune cells offers a valuable strategy to identify gene targets pertinent to cell-based therapeutic interventions».
7. Section 2.3. limitations of current CAR-T cell therapy
The phrase should begin with a capital letter
8. Page 6. Prior studies have highlighted the regulatory influence of EZH1 in maintaining hematopoietic multipotency, where its suppression has been shown to promote lymphoid lineage potential during embryonic development in both mouse and zebrafish models
9. Didn’t find about zebrafish model in this article Vo, L.T., Kinney, M.A., Liu, X., Zhang, Y., Barragan, J., Sousa, P.M., Jha, D.K., Han, A., Cesana, M., Shao, Z., et al. (2018). Regulation of embryonic haematopoietic multipotency by EZH1. Nature 553, 506–510. 10.1038/nature25435. Need more references
10. It isn’t any information about some objects related to the topic of the publication, such as:
· Pluripotent stem cell-derived CAR-macrophage cells: doi: 10.1186/s13045-020-00983-2; doi: 10.1136/jitc-2023-007705.
· Human iPSC-derived CD4+ Treg-like cells doi: 10.1016/j.stem.2024.05.004.
11. Poor information about iPSC-Derived Natural Killer Cells, although there are several works devoted to these important objects: doi: 10.1016/j.stem.2018.06.002.
After correcting these shortcomings, the manuscript can be published.
Reviewer 3 Report
Comments and Suggestions for Authors
1) The authors described the abstract is almost background about CAR-T Cell therapy. The abstract should be revised. One or two sentences are enough. They should add the most important results (especially numerical results) from previous works about iPSC Technology for cancer therapy.
2) I strongly advise the authors to draw a graphical abstract at the end of the introduction.
3) Is it possible to use CAR-T Cell therapy for the treatment of silent tumors? Like triple-negative breast cancer.
4) It is good that the authors mention available commercial technology based on iPSC-derived CAR T-cells and whether this method has been approved by the FDA.
5) Is it possible to use CAR T-Cell and/or iPSC-derived CAR T-Cell in combination therapy (with chemotherapy and/or monoclonal antibody).
Reviewer 4 Report
Comments and Suggestions for Authors
This review article is well structured and correctly documented. The question of conditioning regimens should be addressed also with allogeneic CAR-T (left part) on figure 1. Approches to overcome GVHD like the use of T cell progenitors or cord blood cells exhibiting low T cell alloreactivity are missing and should be discussed. See the review by Marcel van den Brink "allogeneic off-the-shelf CAR T cells : challenges and advances".
Author Response
This review article is well structured and correctly documented. The question of conditioning regimens should be addressed also with allogeneic CAR-T (left part) on figure 1. Approches to overcome GVHD like the use of T cell progenitors or cord blood cells exhibiting low T cell alloreactivity are missing and should be discussed. See the review by Marcel van den Brink "allogeneic off-the-shelf CAR T cells : challenges and advances".
Response: We appreciate the reviewer’s comments. We have revised the figure 1 and provided the following discussion to address the conditioning regimens and approaches to over GvHD.
“It is important to note that host cell-mediated allorejection poses a significant concern for allogeneic CAR-T cell therapy, particularly with respect to host T cell and NK cell-mediated allorejection[37]. Mitigating allorejection is critical for ensuring the long-term sustainability and efficacy of off-the-shelf cell products. Various strategies have been employed to enable allogeneic CAR-T cells to evade host cell recognition, including ablation of HLA molecules, overexpression of NK cell inhibitory ligands, overexpression of anti-apoptotic genes, immunomodulation, utilization of stem cell technology to produce cells with low alloreactivity, and HLA matching along with careful donor selection[5,37,38]. To enhance the persistence and efficacy of allogeneic CAR-T cells, lymphodepletion using agents such as alemtuzumab, fludarabine, and cyclophospha-mide is necessary[39–41].
Reviewer 5 Report
Comments and Suggestions for Authors
It is well known Chimeric Antigen Receptor (CAR) engineered T-cell (CAR-T) therapy is very important approach for cancer treatment.
However, CAR-T cell therapy sometimes face the problems of availability of T cells from the donor or in the patient.
Induced pluripotent stem cell (iPSC) technology will be useful to dissolve the restriction to prepare T-cell for CAR-T therapy.
The comprehensive current progress for combination iPSC and CAR-T are summarized in this review.
This review will surely be very useful for readers in this category.
Although the concepts of the review is quite important, but there are some points in the manuscript that are difficult to understand.
After authors dissolve the problems and make revised manuscript, the manuscript should be accepted.
It is very difficult to read Table 1. The authors should consider to improve readability for the table (example: rotating this table sideways etc.)
Author Response
It is well known Chimeric Antigen Receptor (CAR) engineered T-cell (CAR-T) therapy is very important approach for cancer treatment.
However, CAR-T cell therapy sometimes face the problems of availability of T cells from the donor or in the patient.
Induced pluripotent stem cell (iPSC) technology will be useful to dissolve the restriction to prepare T-cell for CAR-T therapy.
The comprehensive current progress for combination iPSC and CAR-T are summarized in this review.
This review will surely be very useful for readers in this category.
Although the concepts of the review is quite important, but there are some points in the manuscript that are difficult to understand.
After authors dissolve the problems and make revised manuscript, the manuscript should be accepted.
It is very difficult to read Table 1. The authors should consider to improve readability for the table (example: rotating this table sideways etc.)
Response: We thank the reviewer’s positive feedback and suggestions, and we have adjusted the Table 1 to improve the readability.
Round 2
Reviewer 1 Report
Comments and Suggestions for Authors
(Second review)
The review articles titled as "iPSC Technology Revolutionizes CAR-T Cell Therapy for Cancer Treatment “effectively" highlights the potential synergy between CAR-T cell therapy and iPSC technology as a promising strategy to address the limitations of CAR-T cell therapy.
I do believe it’s a topic of high relevance and importance to the readers of the journal and overall scientific community.
However, based on my previous comments the author have significantly imporoved the review article, added new points in discussion that imporved the scientific vigor.
Therefore, I am encouraged to endorse the publication of the review article.
Best wishes…
Reviewer 3 Report
Comments and Suggestions for Authors
The authors answered my comments well, and I recommend publishing the manuscript.